# Dynamic Rounding Stability in Through-Feed Centerless Grinding

**Fukuo Hashimoto** 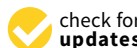

Advanced Finishing Technology Ltd., Akron, OH 44319, USA; fukuohashimoto@gmail.com

**Abstract:** The through-feed method in centerless grinding allows manufacturers to produce cylindrical parts at much higher levels of productivity than can be achieved with in-feed grinding, so it has been extensively employed in industry. However, its rounding mechanism is not yet well understood due to the complexity of the through-feed process. This paper presents the fundamental parameters, such as material removal rates, forces, and so on in the through-feed grinding, and analyses on the grinding system with feedback loops, including regenerative functions and the machine dynamic functions. Further, the characteristic roots of the system, representing the number of waves and the growth rates of the harmonics in roundness, are identified at each grinding position from entry to exit. To evaluate the grinding process stability, a rounding stability index (RSI) was proposed. It was demonstrated that the analytical tool modeled in this paper can identify the optimum operational conditions by the RSI for achieving desired grinding productivity and accuracy. Finally, the model is verified with grinding tests, and the nm-order roundness obtained by the tests is shown.

**Keywords:** grinding; centerless; through-feed grinding

## 1. Introduction

Through-feed centerless grinding has been extensively employed for the mass production of cylindrical parts, such as pins, rods, cylindrical rollers, needle rollers, tapered rollers, and rings, because of the considerably higher productivity it offers compared to in-feed grinding. With through-feed grinding, when the operations are properly set up, it is not unusual to produce hundreds of parts per minute within 0.1 μm-order roundness [1].

Over the last several decades, a great deal of research has been devoted to understanding the rounding mechanism of in-feed centerless grinding, and the theory has been well established [2–7]. However, the setup parameters in through-feed grinding continuously vary at the different grinding positions from entry to exit, so the rounding theory [8] derived under in-feed conditions is not directly applicable to the through-feed process.

In the past, several studies have been conducted to analyze the through-feed process [9–12]. While these studies have made significant progress toward understanding through-feed grinding, the explicit guidance that would be applicable in practice for achieving optimum productivity, and accuracy is not yet available. The current practices for proper setup conditions still largely rely on a cut-and-try method, which becomes a major constraint in automated quick-changeover operations.

This paper describes the geometrical arrangements of the wheels, blade, and workpieces found in through-feed centerless grinding. Fundamental parameters such as material removal rates, grinding forces, maximum production rates, and depth of cut are presented. Then, a newly conceived grinding system with feedback loops including the regenerative functions and machine dynamic functions is introduced, and the characteristic equation of the system as a function of the grinding position is derived. By solving the equation, the characteristic roots, representing the number of waves and the growth rates of the harmonics in roundness, are found. The dynamic rounding stability of the

system is calculated using the roots found at different positions along the grinding length. The root loci are investigated at each grinding position from entry to exit, and a rounding stability index (RSI) is proposed in order to evaluate throcess stability, as well as rounding stability.

To estimate the incoming roundness of the workpieces, a roundness function with harmonic distributions was proposed, and it was applied as the initial roundness of the workpiece for the grinding process simulations. After conducting numerous process simulations under various stable conditions, it was demonstrated that the analytical tool modeled in this paper can identify the optimum operational conditions for achieving desired grinding productivity and accuracy. Finally, the model was verified with grinding tests, and the nm-order roundness obtained by the tests was shown.

## 2. Principles of Through-Feed Centerless Grinding

Figure 1a,b show the geometrical arrangement of the wheels, blade, and workpieces, and the section view at the center of the grinding length, $L_g$, where $\theta$ is the blade angle. The regulating wheel (RW) has a skew angle, $\xi$, to give the workpiece (WP) axial feed, and is trued at a height where RW contacts with the workpieces along the grinding length, causing the RW body to become a hyperboloid with the profile, as shown in Figure 2a,b. As a result, the center-height angle, $\gamma$, representing the center-height, $h$, of WP varies along the grinding positions from the entry to the exit, as shown in Figure 2c.

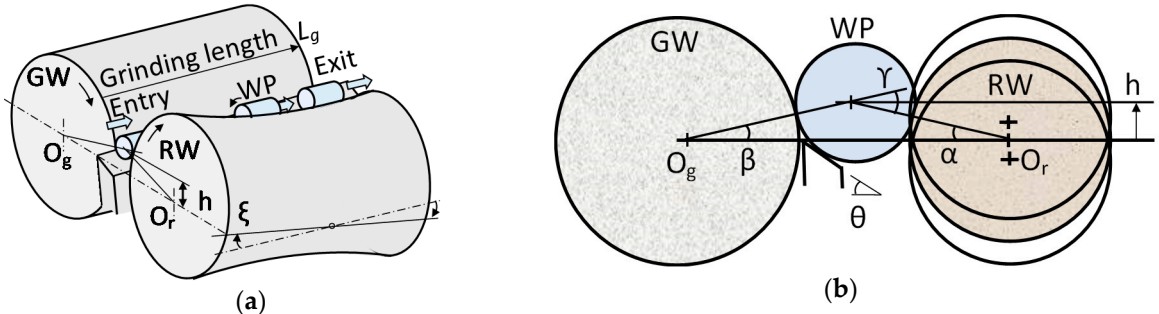

**Figure 1.** Through-feed centerless grinding. (**a**) Geometrical arrangement; (**b**) Section view at the center.

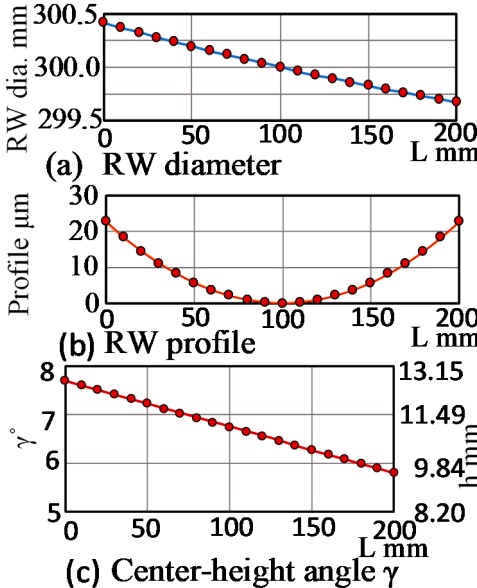

**Figure 2.** Regulating wheel (RW) profile and center-height angle. (**a**) RW diameter; (**b**) RW profile; (**c**) Center-height angle $\gamma$.

As the axial-feed velocity $v_a$, of WP can reprent as ($\pi \cdot d_r \cdot N_r \cdot \sin\xi$), the specific material removal rate, $Q_w'$, can be represented as follows [13]:

$$Q_w' = \frac{\pi^2 \sin\xi d_w d_r N_r S}{2L_g} \tag{1}$$

where $d_w$ is WP diameter, $d_r$ is RW diameter, $N_r$ is rotational speed of RW, and $S$ is the diameter stock removed per pass. The normal grinding force, $F_n$, is provided by:

$$F_n = \eta F_t = \eta u \frac{\pi^2 \sin\xi d_w d_r N_r S}{2v_s} \tag{2}$$

where $F_t$ is the tangential force, $\eta$ is the force ratio ($F_n/F_t$), $u$ is the specific energy, and $v_s$ is the grinding speed. The productivity, $N_p$, in pieces per unit time is given by $v_a/b$, where $b$ is the length of WP. The maximum throughput, $N_{pmax}$, in pieces per unit time is given by:

$$N_{pmax} = \frac{2(P_m - P_i)}{\pi u d_w b S} \tag{3}$$

where $P_m$ is the grinding wheel (GW) motor power, and $P_i$ is the idle power before grinding. The nominal depth of cut, $a$, per revolution of WP can be expressed by:

$$a = \frac{\pi \sin\xi d_w S}{2L_g} \tag{4}$$

Note that the nominal depth of cut is independent of the WP speed, $n_w$, and RW speed, $N_r$.

## 3. Through-Feed Centerless Grinding System

Figure 3 shows the system model representing the dynamic rounding mechanism in through-feed centerless grinding. The system loop consists of the regenerative centering function, $F_L(s)$, and the dynamic compliance function, $G_L(s)$. The input of the function ($F_L(s)$) is the actual approach ($\Delta r$) of GW to WP, and the output is the instantaneous depth of cut ($t_c$) that turns into the machine deformation (de) through the function $G_L(s)$.

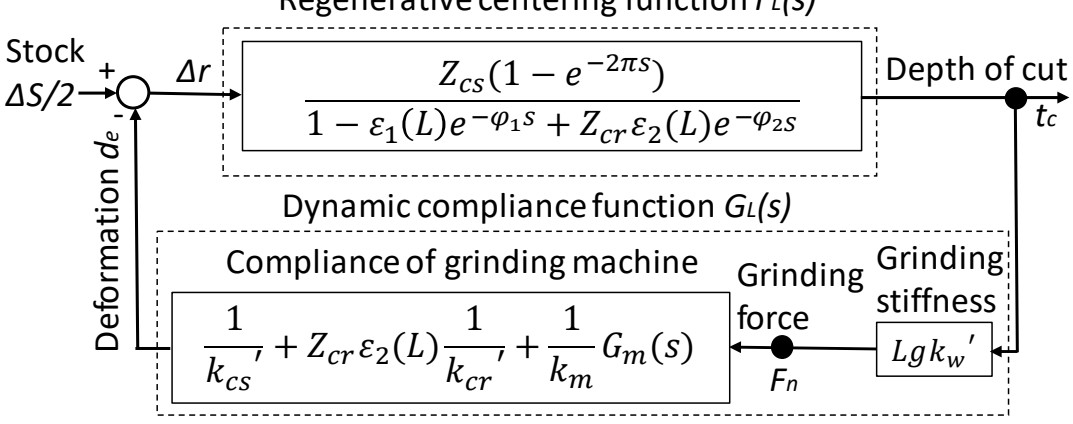

**Figure 3.** Dynamic rounding loop at grinding position in through-feed centerless grinding system.

The characteristic equation of the system at the grinding position, $L$, can be expressed as follows:

$$1 + F_L(s)G_L(s) = 0 \tag{5}$$

$$F_L(s) = \frac{Z_{cs}\left(1 - e^{-2\pi s}\right)}{1 - \varepsilon_1(L)e^{-\varphi_1 s} + Z_{cr}\varepsilon_2(L)e^{-\varphi_2 s}} \tag{6}$$

$$G_L(s) = L_g k_w'\left[\frac{1}{k_{cs}'} + Z_{cr}\varepsilon_2(L)\frac{1}{k_{cr}'} + \frac{1}{k_m}G_m(s)\right] \tag{7}$$

$$\varepsilon_1(L) = \frac{\sin\{\alpha(L) + \beta(L)\}}{\cos\{\theta - \alpha(L)\}} \tag{8}$$

$$\varepsilon_2(L) = \frac{\sin\{\theta + \beta(L)\}}{\cos\{\theta - \alpha(L)\}} \tag{9}$$

where $s$ is a Laplace operator expressed by $s = \sigma + j \times n$. The real part ($\sigma$) is the growth rate of the wave amplitude, and the imaginary part ($n$) is the harmonics in undulation per revolution (UPR). $\phi_1$ and $\phi_2$ are the phase lags to the grinding point from the blade and from the RW, respectively. $\varepsilon_1(L)$ and $\varepsilon_2(L)$ are the feedback factors from the blade and the RW contact points. $\gamma(L)$ is the center-height angle ($\alpha(L)$ + $\beta(L)$) at $L$ (see Figure 1).

These parameters are the functions of the grinding position, $L$. $k_w'$ is the specific grinding stiffness, and $k_{cs}'$ and $k_{cr}'$ are the specific contact stiffnesses of GW and RW. $k_m$ is the static machine stiffness and $G_m(s)$ represents the dynamic compliance of the grinding machine. $Z_{cs}$ and $Z_{cr}$ are the geometrical filtering functions of GW and RW, respectively [14].

Solving the characteristic Equation (5) at each grinding position ($L$) by using the diagrammatical coincidence method described in [14], the characteristic root ($s = \sigma + j \times n$) for a specific harmonic ($n$) UPR can be found at any grinding position ($L$) from the entry to the exit. The amplitude ($An$) of harmonic ($n$) UPR at the grinding time ($t$) can be calculated by the following equation:

$$A_n(t) = A_{n0}e^{2\pi n_w \sigma t} \tag{10}$$

where $A_{n0}$ is the initial amplitude of n UPR in the roundness error, and $n_w$ is the rotational speed of WP in rps. When the growth rate ($\sigma$) is negative ($\sigma < 0$), the amplitude ($A_n$) of the $n$ UPR is reduced and roundness is improved with grinding time ($t$). This root is identified as stable. On the other hand, when $\sigma > 0$, the amplitude of $n$ UPR increases with $t$, and it appears as work-regenerative chatter vibration with deteriorating roundness. Under these conditions, the grinding system becomes unstable. When $\sigma = 0$, no amplitude change occurs during the grinding process.

Figure 4a–c are examples of the characteristic roots found at the entry, the center, and the exit along the grinding length ($L_g$). The distribution patterns of the characteristic roots are identified between even $n$ UPR and odd $n$ UPR. At entry, $L = 0$, there are four unstable roots ($\sigma > 0$) which are 10, 11, 20, and 22 UPRs. The amplitudes of these harmonics grow, particularly the 11 UPR, which rapidly builds up. Additionally, the roots of 3, 5, and 24 UPRs are $\sigma \approx 0$, indicating that these harmonics are difficult to remove. At the center ($L = 100$ mm) and the exit ($L = 200$ mm), the root distributions are different from the entry's, and the values of the growth rates vary at $L$. The growth rates of 20 and 22 UPRs increase when the WP travels toward the exit from the entry.

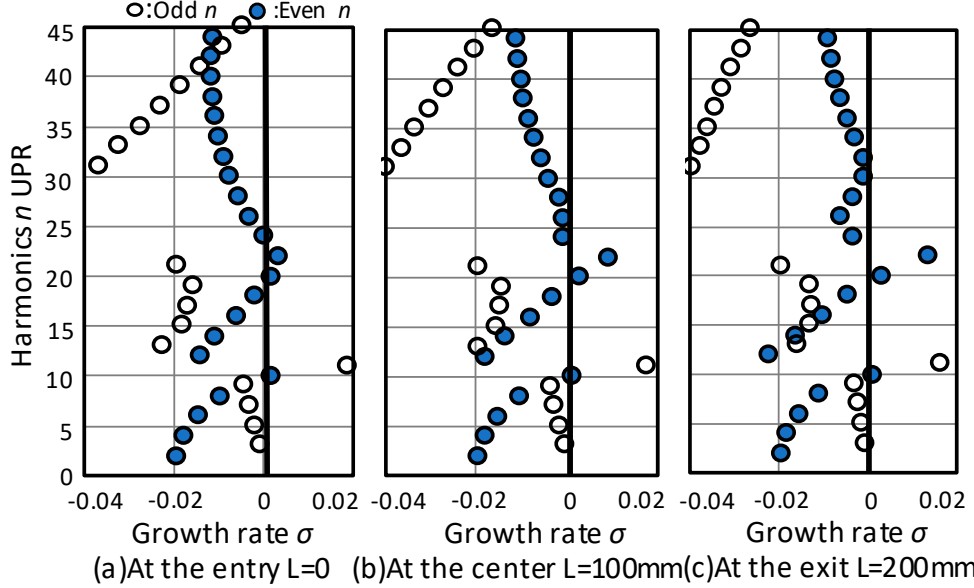

**Figure 4.** Characteristic roots at the different grinding points. (**a**) At the entry L = 0; (**b**) At the center L = 100 mm; (**c**) At the exit L = 200 mm. Conditions: $d_w$ = 25 mm; $I_w$ = 30 mm, $d_s$ = 455; $L_g$ = 200 mm, $d_r$ = 300 mm, $\gamma_0$ = 6.75°, $N_r$ = 50 rpm, $\xi$ = 1.5°, $k_w'$ = 2 kN/mm, $k_{cr}'$ = 3 kN/mm, $k_{cs}'$ = 1 kN/mm, $k_m$ = 60 kN/mm, *fn1* = 100 Hz, *fn2* = 200 Hz, $\zeta 1$ = 0.08, $\zeta 2$ = 0.05, $Z_{cs}$ = 1, $Z_{cr}$ = 1.0, $\varepsilon_1(L)$ = 0.

## 4. Model of Initial Roundness and Dynamic Rounding Stability

### 4.1. Initial Roundness

The out-of-roundness ($r_L(\varphi)$) at the $L$ position is expressed by:

$$r_L(\varphi) = \sum_{n=2}^{\infty} A_{n0}(L) \cos(n\varphi + \tau_n) \tag{11}$$

where $A_{n0}$ is the initial amplitude of $n$ UPR, $\phi$ is the rotation angle, and $\tau_n$ is the phase of n UPR. To investigate roundness stability during the through-feed process, it is essential to determine whether the final roundness after grinding improved from the initial roundness. Although the initial roundness can be obtained by measuring the out-of-roundness of the incoming WP with harmonic analysis, it is very time-consuming to measure the out-of-roundness of all the workpieces. Thus, for simulation purposes, the initial roundness $r_0(\phi)$ was defined by assuming the following initial amplitude, $A_{n0}$:

$$A_{n0}(n) = A_{20} e^{-\frac{(n-2)}{B}} \tag{12}$$

where $A_{20}$ is the initial amplitude of $n$ = 2 UPR at $L$ = 0, and B is a constant. Figure 5a shows the initial amplitude distributions, calculated assuming $A_{20}$ = 1 µm of $n$ = 2 UPR. The intensity of the higher harmonics can be modified by setting the $B$ value. Figure 5b shows the polar plot of roundness errors obtained by Equation (11) with $A_{20}$ = 1 µm and $B$ = 10. The calculated roundness is 4.72 µm.

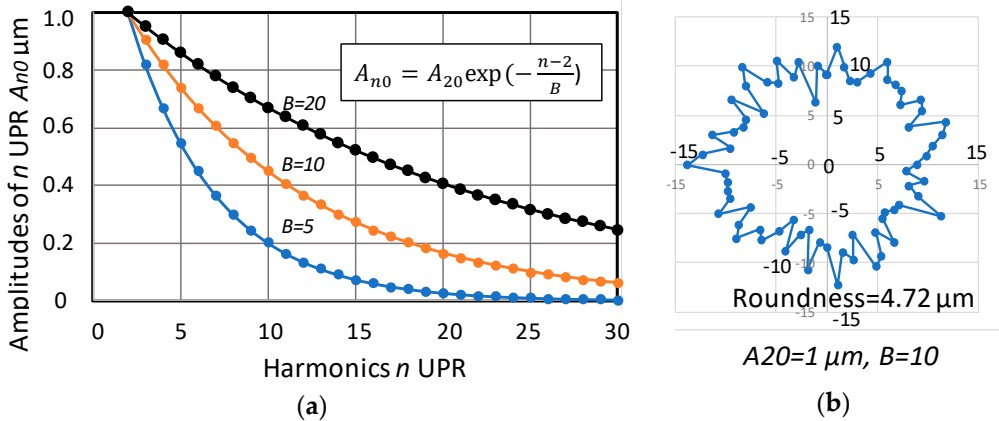

**Figure 5.** Initial amplitude distributions and roundness. (**a**) Initial amplitude distribution; (**b**) Initial roundness.

### 4.2. Dynamic Rounding Stability

A rounding stability index (RSI) is required to assess whether or not the operational setup conditions will provide the intended roundness and productivity level. The rounding stability of the through-feed centerless grinding system depends on the maximum value of the growth rate ($\sigma_{max}$) among all the roots at each grinding position. The series of $\sigma_{max}(L)$ values along the grinding position $L = m\Delta L$, where m is the section number from zero to $N$, and $\Delta L$ is the section increment along $L$. The RSI is defined as follows:

$$RSI = \frac{1}{(N+1)} \sum_{m=0}^{N} 10^3 e^{-(L_g - m\Delta L)/L_g} \cdot \sigma_{max}(m\Delta L) \tag{13}$$

The coefficients of $\sigma_{max}$ in Equation (13) are the influencing factors of the RSI. The $\sigma_{max}$ near the grinding entry has a smaller influence on the *RSI*, and the $\sigma_{max}$ close to the grinding exit has greater influence. Figure 6a,b showed the maximum growth rate ($\sigma_{max}$) at each grinding position and the $\sigma_{max}$ multiplied by the influencing factors. The conditions for the RSI calculations are the same as those shown in Figure 4.

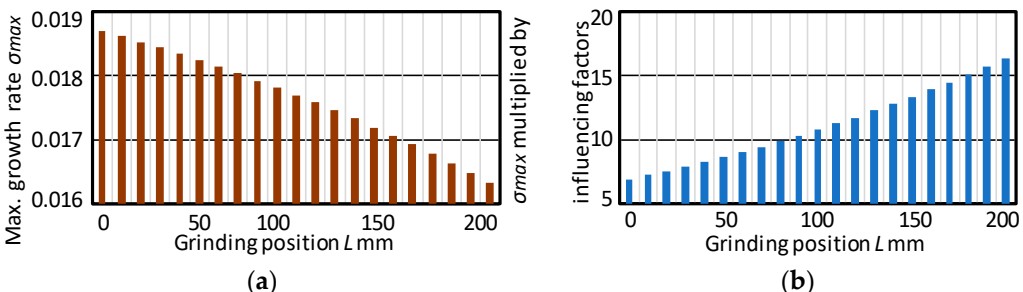

**Figure 6.** RSI calculation. RSI = 11.1. (**a**) Max. growth rate $\sigma_{max}$; (**b**) $\sigma_{max}$ multiplied by influencing factors. Conditions: $d_w$ = 25 mm, $b$ = 30 mm; $d_r$ = 300 mm; $\gamma_0$ = 6.75°, $N_r$ = 50 rpm, $L_g$ = 200 mm, $\Delta L$ = 10 mm, $N$ = 20.

The RSI obtained in this manner is 11.1, which is the average of the values shown in Figure 6b. A value of RSI > 0 indicates that the rounding process is unstable.

## 5. Simulations and Validation

The machine specifications and the grinding conditions used for the simulations are shown on Table 1. The maximum amplitude of machine vibration is limited to the nominal depth of cut $a$ calculated by Equation (4).

**Table 1.** Machine specifications and grinding conditions.

| Parameter | Symbol | Value |
|---|---|---|
| Grinding wheel diameter | $d_s$ | 455 mm |
| Regulating wheel diameter | $d_r$ | 300 mm |
| Grinding length | $L_g$ | 200 mm |
| Skew angle of regulating wheel spindle | $\xi$ | 1.5° |
| Grinding speed | $v_s$ | 45 m/s |
| Grinding wheel motor power | $P_g$ | 30 kW |
| Grinding wheel motor idling power | $P_i$ | 5 kW |
| Workpiece: cylindrical roller diameter | $d_w$ | 15 mm |
| Workpiece length | $b$ | 18 mm |
| Grinding stocks in diameter | $S$ | 0.25 mm |
| Nominal depth of cut | $a$ | 0.77 μm/rev |
| Specific grinding energy | $u$ | 50 J/mm$^3$ |
| Force ratio ($F_n/F_t$) | $\eta$ | 2.0 |
| Maximum throughputs | $N_{pmax}$ | 206 pieces/min |
| 1st natural frequency of grinding machine | $fn1$ | 100 Hz |
| Damping factor of 1st natural frequency | $\zeta1$ | 0.05 |
| 2nd natural frequency of grinding machine | $fn2$ | 200 Hz |
| Damping factor of 2nd natural frequency | $\zeta2$ | 0.08 |

*5.1. Simulation I*

The conditions for simulation I are:

- Rounding stability index (RSI) = 3.07,
- RW rotational speed ($N_r$) = 26 rpm,
- WP rotational speed ($n_w$) = 8.67 rps,
- Throughputs ($N_p$) = 35.6 pcs/min,
- Specific material removal rate ($Q_w'$) = 0.31 mm$^3$/mm·s, and
- Normal grinding force ($F_n$) = 141 N.

There are two unstable roots (24 UPR and 26 UPR) caused by *fn2* = 200 Hz, as shown in Figure 7a. The positive growth rates are first reduced, and then increased toward the exit, as shown in Figure 7b. The roundness changes at L, and the final roundness ends up at 2.53 μm, with a beat undulation of 24 and 26 UPR (see Figure 7c,d). As predicted by the RSI, the rounding stability is considered unstable.

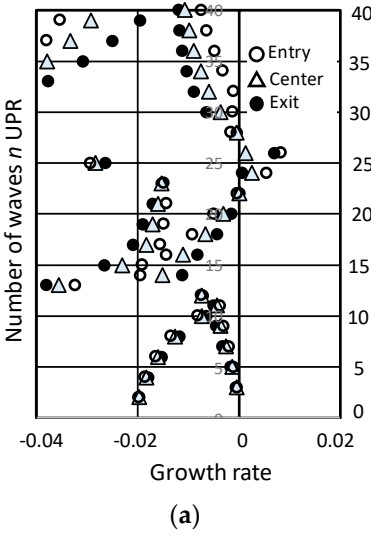

(**a**)

**Figure 7.** *Cont.*

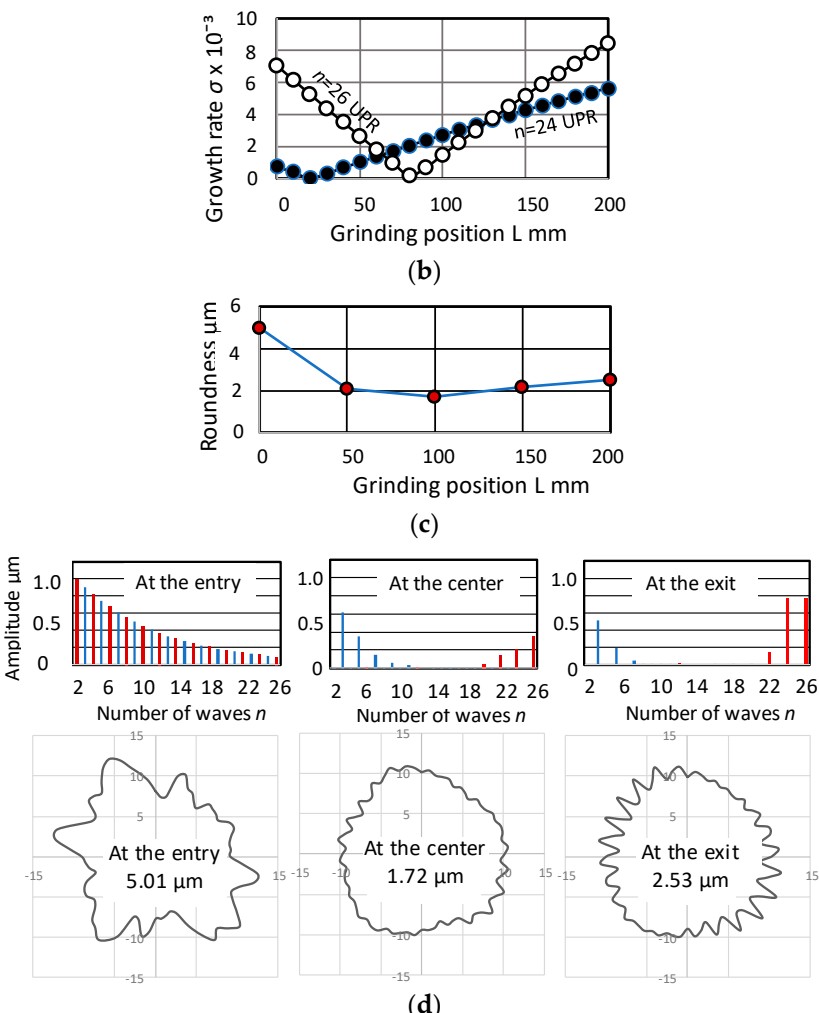

**Figure 7.** Simulation I: Unstable process RSI = 3.07. (**a**) Characteristic root distributions; (**b**) Growth rates of 24 UPR and 26 UPR; (**c**) Roundness vs. grinding positions; (**d**) Progress of roundness.

*5.2. Simulation II*

The conditions for simulation II are:

- Rounding stability index (RSI) = 7.79,
- RW rotational speed ($N_r$) = 40 rpm,
- WP rotational speed ($n_w$) = 13.3 rps,
- Throughputs ($N_p$) = 54.8 pcs/min,
- Specific material removal rate ($Q_w'$) = 0.48 mm$^3$/mm·s, and
- Normal grinding force ($F_n$) = 217 N.

In this case, only one unstable root (16 UPR) appears (see Figure 8a). The harmonic ($n$) = 16 UPR generates 213 Hz ($n \times n_w = 16 \times 13.3$), which resonates with the second-order natural frequency (*fn2* = 200 Hz) of the machine. As a result, the final roundness ends up at 2.04 μm with 16 lobes, as shown in Figure 8b,c. The through-feed grinding process induces work-regenerative chatter vibration, and the rounding stability is unstable, as the RSI value indicates.

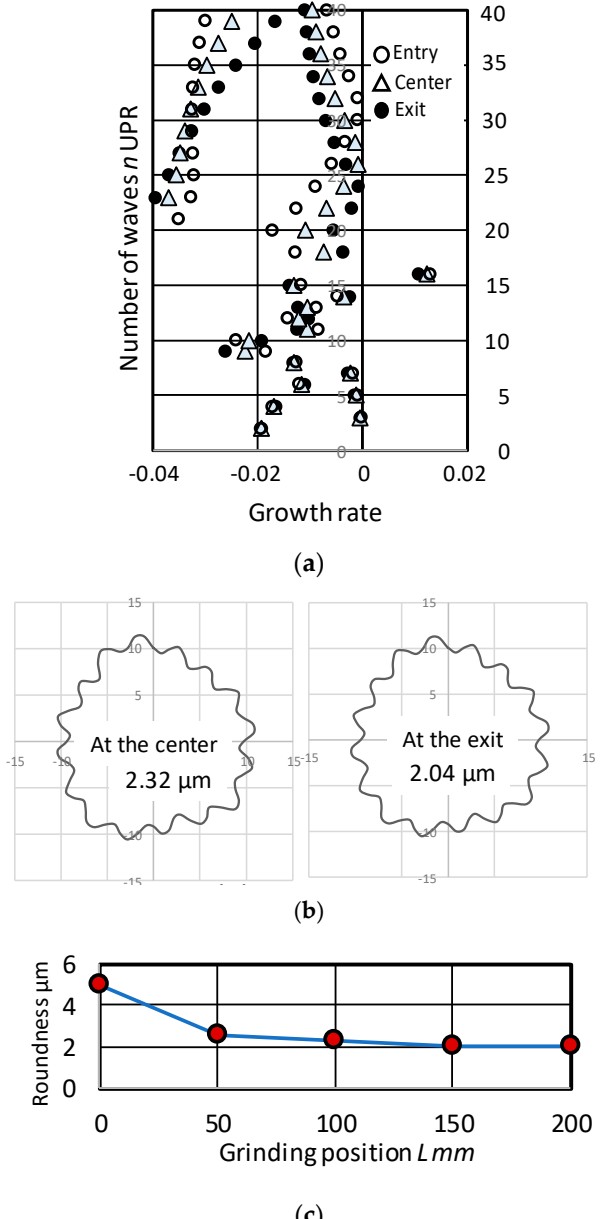

**Figure 8.** Simulation II: Unstable process, RSI = 7.79. (**a**) Characteristics roots; (**b**) Roundness; (**c**) Progress of roundness.

*5.3. Simulation III*

The conditions for simulation III are:

- Rounding stability index (RSI) = −0.33,
- RW rotational speed ($N_r$) = 120 rpm,
- WP rotational speed ($n_w$) = 40.0 rps,
- Throughputs ($N_p$) = 164.3 pcs/min,
- Specific material removal rate ($Q_w'$) = 1.45 mm³/mm·s, and
- Normal grinding force ($F_n$) = 650 N.

As shown in Figure 9a, in Simulation III there is no unstable root at any section along the grinding length. During the first grinding from the entry to $L$ = 50 mm, the incoming roundness (5.01 μm) is rapidly reduced to 0.67 μm. Then, the roundness is further improved and finally reaches 0.03 μm.

When a negative value of RSI is achieved, the rounding stability is dynamically stable. Dynamically stable rounding is expected to significantly improve roundness, as shown in Figure 9b,c.

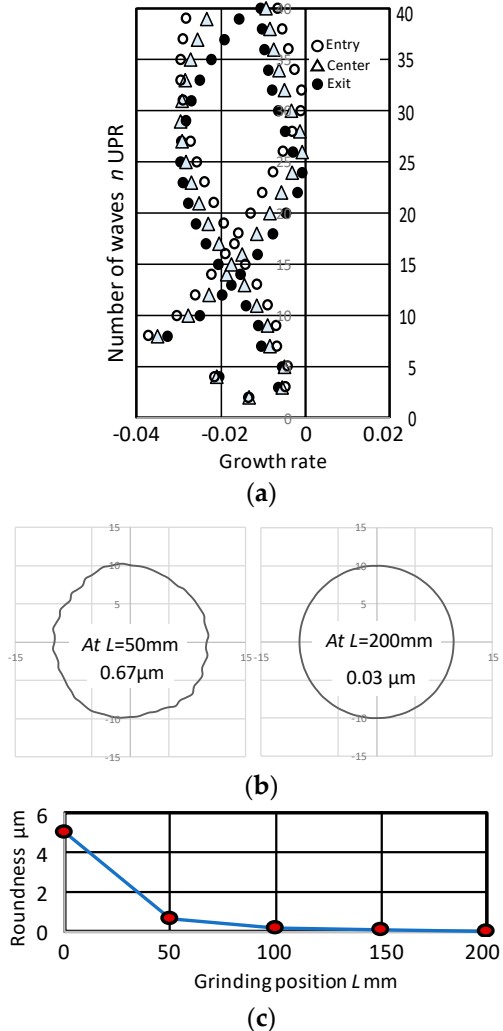

**Figure 9.** Simulation III: Stable process, RSI = −0.33. (**a**) Characteristics roots; (**b**) Roundness; (**c**) Progress of roundness.

*5.4. Validation*

As discussed above, the RSI represents not only the stability criterion of the grinding system, but also a degree of rounding stability. Figure 10 shows the relationship between RSI and the final roundness in simulations. The roundness tends to be greater with increased RSI, and also tends to become greater when the WP diameter is larger. When a negative RSI is reached, the roundness errors can be minimized, and the system's stability maintained.

The grinding tests were conducted to verify the results of the simulations. For this through-feed grinding test, a new centerless grinding machine was designed and built [15,16]. The developed machine has one-order higher dynamic stiffness and motion accuracy than a conventional machine. Figure 11 shows the roundness chart obtained by the experimental tests under the conditions of Simulation III. The through-feed centerless grinding processes were very stable without any vibration, as predicted in the simulation III. The obtained roundness of 0.026 μm has good agreement with the predicted 0.03 μm (see Figure 9b) and has the highest accuracy of the results reported so far [9–11].

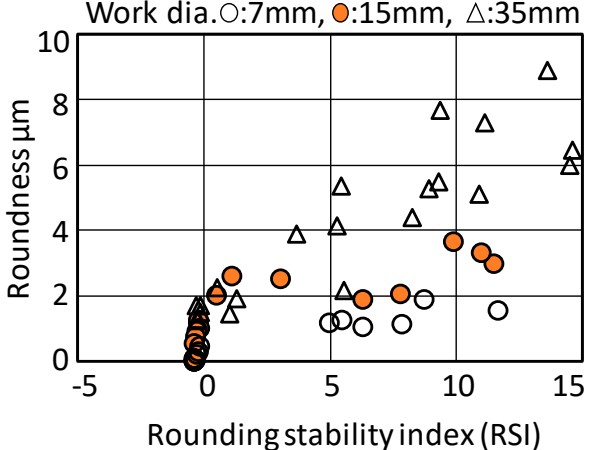

**Figure 10.** RSI vs. roundness (simulation).

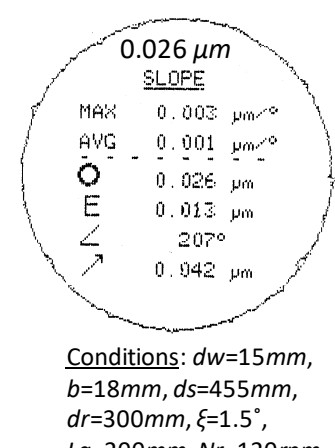

Conditions: *dw*=15*mm*,
*b*=18*mm*, *ds*=455*mm*,
*dr*=300*mm*, *ξ*=1.5˚,
*Lg*=200*mm*, *Nr*=120*rpm*

**Figure 11.** Roundness (experimental).

## 6. Conclusions

In through-feed centerless grinding, this paper presents the fundamental grinding parameters and analyses on the grinding system with feedback loops including regenerative functions and the machine dynamic functions. The characteristic equation of the grinding system was derived and the roots, representing the number of waves and the growth rates of harmonics in roundness, were found. The root loci were investigated along the grinding length, and a rounding stability index (RSI) was proposed in order to evaluate the through-feed process stability as well as rounding stability. The results of through-feed process simulations with different RSI conditions were discussed. It was demonstrated that the analytical tool modeled in this paper can identify the optimum operational conditions for achieving the desired grinding productivity and accuracy. Finally, the model was verified with through-feed grinding tests. The following conclusions were obtained.

(1) The principles of through-feed centerless grinding were described, and the fundamental parameters, such as the material removal rates, grinding forces, maximum production rates, and depth of cut were presented.

(2) A through-feed grinding system dependent upon the grinding positions was deduced, and the characteristic equation is found. Solving the equation reveals the characteristic roots, which allow us to calculate the transient changing processes in roundness along the grinding position from entry to exit.

(3)  To estimate the incoming roundness of the workpieces, a roundness function with harmonic distributions was proposed. It was applied as the initial roundness of the workpiece for grinding process simulations.

(4)  The rounding mechanisms in through-feed grinding were discussed, and the RSI to define system stability as well as optimum setup conditions were proposed.

(5)  Simulations of the proposed model for through-feed grinding were conducted, which showed that stable conditions can be obtained by achieving a negative RSI.

(6)  The model developed here can provide not only which harmonics were built up or converged, but also the final roundness. Additionally, it was capable of eliminating the cut-and-try method in setup operations and giving the optimum setup conditions that improve the grinding accuracy and the productivity.

(7)  The model was verified with through-feed centerless grinding tests, and the nm-order roundness obtained by the tests was shown.

**Funding:** This research received no external funding.

**Conflicts of Interest:** The authors declare no conflict of interest.

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
