# Peer review of "Dynamic Rounding Stability in Through-Feed Centerless Grinding"

_inventions, doi:10.3390/inventions5020017_

Round 1
Reviewer 1 Report
A highly competent review of the dynamics of centerless grinding conducted by a well-known expert in the field of grinding.
Author Response
To Reviewer,
Thank you so much for taking time out of your busy schedule.
I appreciate your review of the paper.
Fukuo Hashimoto
Reviewer 2 Report
This paper describes a through-feed grinding system dependent upon the grinding positions, and revealed the characteristic roots along the grinding position from the entry to the exit. And the dynamic rounding stability of the system is calculated using the characteristic roots found at different positions along the grinding length. Through the process simulations under stable conditions, the author showed the analytical model can identify the optimum operational conditions for achieving desired grinding productivity and accuracy.
Based on the author's long life experience of research with respect to the stability of centerless grinding, this paper discusses the rounding mechanisms in through-feed grinding well, and proposes a stability index (RSI) to define system stability.
Those results could be applied in the field of through-feed grinding.
Reviewer agrees to be published.
Author Response
To Reviewer,
Thank you so much for the review and taking time out of your busy schedule,
I appreciate your comments and well understanding of the complex issues and analytical methods.
Best Regards,
Fukuo Hashimoto
Reviewer 3 Report
Reviewed article is very interesting and write at high scientific level. The subject is very important and not often developed by the researches, thus submitted manuscript have a great value and broaden the knowledge on centerless grinding. Presentation method is good and in accordance with generally accepted standards in that area. Figures, tables as well as terminology are mostly clear and precise. Described method was correctly verified and compared with standard approach to this problem. Below are listed some substantive remarks that should be taken into consideration by the Author to improve reviewed text:
- the abstract should include information about: new methods, results, concepts, and conclusions – in its current form, the abstract needs to be rewritten to include more information on achievements described in the manuscript;
- literature review should be improved providing more references to recent works from the area of described study;
- at the end of the introduction should be clearly and concise given the research gap to create the appropriate lead up for the motivation of the work;
- the Author should more carefully and more detailed describe methodology of validation;
- I suggest to provide more precise information about used measurement positions;
- the machine specifications and the grinding conditions given at the begging of section 5 should be rewritten in the form of table – also parameters of simulations I-II should be given in tables;
- in the discussion section more references to already known results from literature should be given;
- the strengths and limitations of the obtained results and applied methods should be clearly described;
- I suggest also to give wider description of potential use of presented findings in scientific research as well as in industrial practice;
- in section 5 lack of analysis with regard to the basic phenomena in the grinding process;
- in conclusions deeper explanation of observed phenomena and trends should be given (conclusions should refer not only to results but also to causes of obtained results) and refer to specific values (results of analysis),
- the conclusion should be improved in term of the new knowledge gained during analysis, which should be concise with the journal scope.
After a careful study of the text sent for review, many editorial comments also come to mind:
- all mathematical/physical symbols should be consequently write italics and proper subscript/superscript notation for better readability of the text;
- many of included figures have more than one element (for example figure 1 have 2 elements: a, and b) – each of the elements should be described in the figure caption;
- lack of punctuation marks after equitation (equitation is part of a sentence);
- consequently all values should be writing with space before its unit (with very few exceptions);
- figure is not a part of the sentence – Fig. 1 and 2 should be placed outside the sentence.
Author Response
- Abstract: The abstract is rewritten.
- End of Introduction: Rewritten
- Section 5.4 Validation: Rewritten
- Conclusions: Rewritten
Other comments:
-References: 2 references are added. Reference papers were reviewed in my CIRP keynote paper [1] and I don't want to repeat them.
-Space before unit: The spaces were added.
-Figures: Each element in the figures has the own caption.
-Each equation has the punctuation. That's why, "where" is used right after the equation. This is a typical style, I believe.
-Experimental results: In industry, the practical information in through-feed centerless grinding is high confidential matter and limited in the disclosure. This paper focused on the description of analytical parameters.
-This is the best the author can do.
Thank you for the review and comments
Regards,
Fukuo Hashimoto

Round 2
Reviewer 3 Report
Most of my comment were taken into consideration by the Author.